# Integrating Systems Thinking and Behavioural Science

**DOI:** 10.3390/bs15040403

**Published:** 2025-03-21

**Authors:** John A. Parkinson, Ashley Gould, Nicky Knowles, Jonathan West, Andrew M. Goodman

**Affiliations:** 1Wales Centre for Behaviour Change, Department of Psychology, Bangor University, Gwynedd LL57 2AS, UK; 2Behavioural Science Unit, Public Health Wales, Cardiff CF10 4BZ, UK; ashley.gould@wales.nhs.uk (A.G.); nicky.knowles@wales.nhs.uk (N.K.); jonathan.west@wales.nhs.uk (J.W.); 3Wales Centre for Behaviour Change, School of Computer Science and Engineering, Bangor University, Gwynedd LL57 2AS, UK; andy.goodman@bangor.ac.uk

**Keywords:** behaviour, change, systems, complexity, design thinking, public health, dual-process, nudge, multidisciplinary

## Abstract

Traditional approaches to changing health behaviours have primarily focused on education and raising awareness, assuming that increased knowledge leads to better decisions. However, evidence suggests these methods often fail to result in sustained behavioural change. The dual-process theory of decision-making highlights that much of our behaviour is driven by automatic, intuitive processes, which educational interventions typically overlook. Compounding this challenge, behavioural research is often conducted on small groups, making it difficult to scale insights into broader societal issues, where behaviour is influenced by complex, interconnected factors. This review advocates for integrating behavioural science with systems approaches (including systems thinking and approaches to complex adaptive systems) as a more effective approach to resolving complex societal issues, such as public health, sustainability, and social equity. Behavioural science provides insights into individual decision-making, while systems approaches offer ways of understanding, and working with, the dynamic interactions and feedback loops within complex systems. The review explores the commonalities and differences between these two approaches, highlighting areas where they complement one another. Design thinking is identified as a useful structure for bridging behavioural science and systems thinking, enabling a more holistic approach to problem-solving. Though some ideological challenges remain, the potential for creating more effective, scalable solutions is significant. By leveraging the strengths of both behavioural science and systems thinking, one can create more comprehensive strategies to address the “wicked problems” that shape societal health and well-being.

## 1. Summary

Traditional approaches to changing behaviours have predominantly relied on educational initiatives and raising awareness. These strategies, grounded in the assumption that increased knowledge will naturally lead to better health decisions, have not been particularly successful. Evidence suggests that simply providing information often fails to translate into sustained behaviour change. This limitation can be partially explained by the dual-process nature of human decision-making. According to dual-process theories, our behaviour is influenced by at least two distinct processes: a fast, automatic, and intuitive process and a slow and deliberative cognitive process. Health interventions, for example, have focused on education and typically engage the latter, neglecting the significant role of the former, which is often more influential than we acknowledge in daily health-related decisions.

Historically, behavioural research has tended to be conducted on small groups or individuals, producing insights that are difficult to generalise, or to scale to larger populations. Public health policies, however, aim to effect change at the societal level, where behaviour is influenced by myriad interconnected factors. This discrepancy highlights a critical gap, or at least benefit realisation potential, in our approach to public health: the need to understand and address health behaviours within the context of complex adaptive systems. Unlike linear processes, complex systems are characterised by numerous interacting components and feedback loops, which produce emergent behaviours that cannot be easily predicted or systematically influenced.

To address these challenges, one promising approach is to integrate behavioural science with systems approaches. Following Jackson (Introduction, page xviii) ([34]), this review uses the term ‘systems thinking’ as a catch-all for all approaches to the understanding and manipulation of complex systems (e.g., critical systems thinking, system dynamics, hard vs. soft systems, etc.). Further distinctions and examples are provided where instructive. Systems thinking provides a framework for understanding the dynamic and interconnected nature of societal behaviours, allowing us to identify leverage points for effective intervention. More broadly, working with systems is about understanding inter-relationships, boundary judgements, and multiple perspectives, in order to identify improvements which are systemically desirable and culturally feasible ([26]; [33]; [56]). By considering the broader system in which behaviours occur, we can develop more robust and sustainable strategies for behaviour change. This integrated approach can better account for the complexities of human behaviour, along with the multifaceted nature of societal influences, ultimately leading to more effective public health interventions and a greater propensity of healthy behaviours.

In summary, the limited success of traditional health behaviour change methods underscores the need for a paradigm shift. By combining insights from behavioural science with systems thinking, we can create more holistic and impactful strategies to improve public health outcomes on a large scale.

## 2. What Is Behavioural Science

Behavioural science is a multidisciplinary field that explores and analyses the ways individuals behave, both in isolation and within social contexts and differing environments and operating contexts. It combines insights from various disciplines such as psychology, sociology, economics, anthropology, and neuroscience to study human behaviour. Behavioural scientists seek to understand the factors driving behaviour, including cognitive processes, social interactions, cultural influences, and environmental triggers—and hence develop theories, design or adapt interventions, and inform policy. The core outcome of interest is usually focused at an individual or group level but can be population wide.

Based on the seminal work of Pavlov and Skinner, behaviour was initially believed to be simply a learnt response to a triggering stimulus. Today, this stimulus-response learning tends to equate to how we consider habits to develop: the repetition of a certain set of actions (response) in a certain context (stimulus) leads to skill development and the establishment of habits, or sensorimotor routines ([54]). A second behavioural learning process is also well understood: that of intended actions leading to the achievement of goals or outcomes. This action–outcome learning differs from automatically triggered habits as they are consciously intentional when created, and likely involve flexibility and dynamic change in order to be completed successfully ([17]; [19]). Today, this tends to equate to our conception of intentional behaviour, based on what cognitive psychologists would call declarative knowledge and working memory. The distinction between these two separable brain processes (e.g., declarative vs. procedural or actions vs. habits) is well acknowledged. Contemporary research focuses on the more granular details of how these processes work, interact, and compete for behavioural control ([42]). There is now a family of dual-process theories of behaviour which also include complementary neuroscience evidence about how multiple brain systems contribute to different types of processing and behaviour ([23]; [24]; [37]; [63]).

An essential principle of these contemporary models is the recognition that human behaviour is influenced by multiple factors. The ‘behavioral insights’ approach is significant. It emerged from observations finding that unconscious drivers, such as social norms, habits, and contextual factors (including physical environment), can sometimes override conscious intentions. This phenomenon, known as the ‘intention-action gap’, is well-documented in the literature, with roots traceable to ancient writings such as Plato’s Republic. The dual-process approach proposes pathways for behavioural control ([22]) which generally work in tandem to guide and energise behaviour. However, certain situations can cause a divergence in processing, leading to conflicts in behavioural control. Type 1 processes, which are phylogenetically older and more directly linked to behavioural control in neural terms, tend to dominate. A prominent example of an unconscious driver of human behaviour is social norms ([10]; [55]). Individuals often choose behaviours by observing others, especially those who share similar characteristics. The dynamics of social norms have been extensively studied ([4]; [10]; [28]; [55]), providing explanations for group behaviour and insights into influencing individual behaviour in desired directions.

Likewise, a parallel contemporary approach to understanding human behaviour is known as the COM-B model, a comprehensive framework that complements the dual-process theory by exploring the higher-level factors that influence behaviour ([47]). COM-B stands for Capability, Opportunity, and Motivation, which together determine behaviour. Capability captures an individual’s psychological and physical capacity to engage in a behaviour (such as having the necessary knowledge and skills). Within the dual-process framework, capability can involve both automatic skills (type 1) and learned abilities that require conscious effort (type 2). Opportunity encompasses external factors that make a behaviour possible or prompt it. This includes physical opportunities (e.g., availability of resources) and social opportunities (e.g., cultural norms, social support). Similarly, opportunities can trigger automatic responses (such as through physical environmental features, or habitual behavioural cues) or be considered in deliberative planning. Motivation involves the processes that energise and direct behaviour. It includes both automatic, emotional drives (type 1) and reflective, goal-oriented considerations (type 2). Motivation can be intrinsic (driven by internal rewards) or extrinsic (driven by external rewards).

Integrating the dual-process theory with the COM-B model provides a robust framework for understanding and influencing behaviour. By recognising the interplay between automatic and controlled processes (Type 1 and Type 2) and the components of capability, opportunity, and motivation, this integrated approach can inform the design of interventions and policies aimed at behaviour change. For instance, a health intervention might enhance capability by providing education (engaging Type 2 processes), create opportunities by making healthy options more accessible (harnessing primarily Type 1 processes), and boost motivation through persuasive messaging that is tailored to the target audience as well as appealing to both emotional and rational aspects (engaging both systems). What these approaches often lack is a consideration of the broader system, whether this be a social group, organisational, societal, or policy-focused ([7]).

## 3. What Is Systems Thinking?

Working with systems captures a wide range of approaches and methods interested in understanding and influencing complex adaptive systems by considering the concepts, interactions, and relationships among various components within that system. Instead of isolating individual elements, systems approaches view problems as part of a dynamic and interconnected whole, which often emerge as a result of unintended interactions amongst the elements. This involves analysing how different components or variables within a system influence one another and how these interactions contribute to the overall behaviour of the system. A focus on the complexity and understanding of systems has roots in Eastern and Western thinking ([16]; [38]; [59]) and has led to seminal frameworks ([2]; [15]; [59]), as well as more focused modelling approaches to understanding organisational, technical ([32]; [34]; [46]; [60]), and human social systems ([9]; [33]; [41]; [56]), also see Jackson ([34]). It has subsequently been applied more broadly to human complex adaptive systems, including health systems, in order to map out and influence core elements such as system rules, actors, inter-relationships, and processes within a system ([4]; [5]; [26]; [32]; [46]; [49]; [60]).

Interactions between human actors within a particular operating context or environment become key elements in understanding and changing systems. A critical consequence of acknowledging dual-process behaviour theory is to see that the majority of human behaviour is not necessarily created intentionally by cognitive decision-making processes, but instead by stimuli in the environment, be they architectural features, other individuals, or contextual cues such as advertising boards or phone notifications. From a systems perspective, this acknowledges that humans are indeed open complex systems in themselves; from a behavioural perspective, it recognises the pre-eminence of the environment and operating context (Type 1 processes) within which behaviour is determined ([2]). Indeed, in her seminal book, ‘Thinking in Systems’ Donella Meadows identified this interaction, on page one, as being the key characteristic of a system: “Once we see the relationship between structure and behavior, we can begin to understand how systems work, what makes them produce poor results, and how to shift them into better behavior patterns.” ([46]). This is the primary and over-riding reason why integrating systems thinking and behavioural science is required in order to address societies greater ills.

Addressing the world’s severe challenges—such as climate change, poverty, health disparities, and global conflicts—requires a fundamental shift in our approach. These issues are inherently complex, multifaceted, and resistant to straightforward solutions. Traditional linear methods, which often focus on isolated aspects of a problematic situation, are insufficient for tackling such deep-rooted and interconnected challenges. Instead, a systems thinking approach offers a more effective framework for understanding and addressing complex problems. For example, Frame et al. integrated a behavioural science and systems thinking approach in creating a ‘Systems change and capability’ team within their Delivery unit of the Ministry for the Environment, in order to redesign environmental policies in the Aotearoa region of New Zealand. Due to the acknowledged complexity of the work the ministry carries out, the team explicitly adopted a systems approach, particularly given multiple perspectives, world views, and language use within the population. They argue for “… emphasis on place-based systems-wide approaches that bridge research and practice rather than adoption of specific tools and processes” (page 3) in tackling complex challenges ([25]). Likewise, [1] ([1]) studied the importance of identifying ‘leverage points’ within a system in order for interventions to have an optimal impact ([45]). Focusing on the transformation of sustainability behaviours, they argued that systems thinking, and in this case the use of leverage points, increases the likelihood of real-world interventions in achieving their aims.

Increasingly, there have been calls to consider the larger system within behavioural science, such as drawing a distinction between individual-framed interventions (i-frame) and system-framed interventions (s-frame) ([7]). Or, alternatively, considering a more continuous approach of placing interventions along a ladder of intervention levels of regulation or control ([52]). In acknowledging the potential for behaviour theory to inform complex challenges, Parkinson et al. developed a model to integrate behavioural psychology, complexity theory, and design thinking ([53]). More recently Hallsworth has published a manifesto for public-health related behavioural science, and within it a call for a better integration of systems thinking and behavioural science ([30], [31]). It is important to acknowledge that there have already been attempts to integrate these approaches ([67]), though it still remains to emerge into the mainstream.

The challenges and the value of integrating behavioural science and systems thinking become self-evident. The overarching goal of systems thinking is moving toward an increased understanding of a systems’ complexity, and in some way influence elements in order to nudge the system into a more desired state. Change might reflect processes or acts within a system, or might reflect a change in the overall system (such as a change in rules) ([33]). For example, if we are interested in public health and obesity, then we might have a personal interest in maintaining our own healthy lifestyle and diet (an individual focus of behavioural science), but we will be much more excited by understanding how we can change the system so that the population becomes healthier (a systems approach to public health of our society). The problem is that as complexity grows, the relationships between causes and consequence break down, i.e., it becomes much less linear and much less predictable. Indeed, the application of a well-intended individual behaviour change intervention to a larger group might even cause unintended consequences that undermine the effort. In other words, we cannot just take a selection of individual-focused behaviour change interventions and then scale them up to a larger system level. We need to understand how the system works and how we might utilise it in our design. An interesting analogy is provided by [13] ([13]), who considered traditional approaches to workplace safety in the US. On the one hand, they identified an approach emphasising behaviour-based safety methods which focus on individual safety behaviours, whilst on the other, they identified culture change approaches focused on higher-level values, trust, and group cohesion. These have historically been seen as separate and mutually exclusive approaches, but the authors concluded that the approaches are in fact complementary and should be integrated to provide a more holistic and coherent approach to safety in the workplace. In an equivalent manner, the integrations of behavioural science and systems thinking provides a complementary and more holistic approach to human systems.

As is found in the broad church of behavioural science, there are also differences in philosophy, theory, and methodology in systems thinking. A common contemporary distinction is between ‘hard’ and ‘soft’ systems ([9]), where the former relates to viewing a system as something that can be engineered or directly modelled (e.g., mathematical), and hence controlled, whereas the latter captures the ‘messiness’ of social systems involving complex human interplay. This distinction also reflects the underlying philosophy and approaches to working with systems. For example, the hard systems approach tends to align with an ontological and systematic approach, trying to understand and categorise a system as a ‘thing’ out there in the world waiting to be understood and manipulated ([26]; [33]; [56]). In contrast, soft system approaches are underpinned by a systemic and epistemological view of building a ‘map of reality’ through experience from within the system itself. Such systems tend to be open and messy ([41]). [56] ([56]) uses the terms duality adopting systemic sensibilities (epistemological) and developing systems thinking literacy (ontological), and whilst some have seen by some as mutually exclusive, they can also be considered as different perspectives that are both valuable for understanding a complex system in certain circumstances. For example, hard systems may exist within a broader soft system, demonstrating that both perspectives can add value to understanding and working with the system ([26]). For these authors, working with Public Health, the ‘soft’ systemic view perhaps provides a more authentic journey, where the world is problematic, linear logic does not apply, and there are no final answers—the inquiry never ends. In this sense, Ison’s distinction of “I spy systems that I can engineer” vs. “I see a messy situation and I can use systems as learning devices” aligns well public health challenges ([26]; [33]).

## 4. Characteristics of Complex Systems

The systemic approach tends to use the inter-relationships, perspectives, and boundaries structure to provide a high-level approach to characterising a complex system, but others are used (e.g., the critical systems thinking approach, encompassing perspectives such as cultural, societal, machine, etc., ([34]). Here, we identify a few overarching characteristics and themes which are of value when integrating with behavioural science. Systems thinking works within complex systems, and theorists have explored the nature of complexity in its own right. A basic understanding of complexity theory is of value here (See Box 1 for one example of approaching complex systems ([41])).

Box 1Working with complexity.When working with systems, it is important to acknowledge that it becomes more difficult to predict causality as a system becomes more complex. This can be schematised as below (Image captured from https://en.wikipedia.org/wiki/Cynefin_framework 17 March 2025), original cynefin model and text ([41]). The figure shows four stages of growing system complexity starting from the simplest ‘clear’ form and ending with ‘chaotic’!:

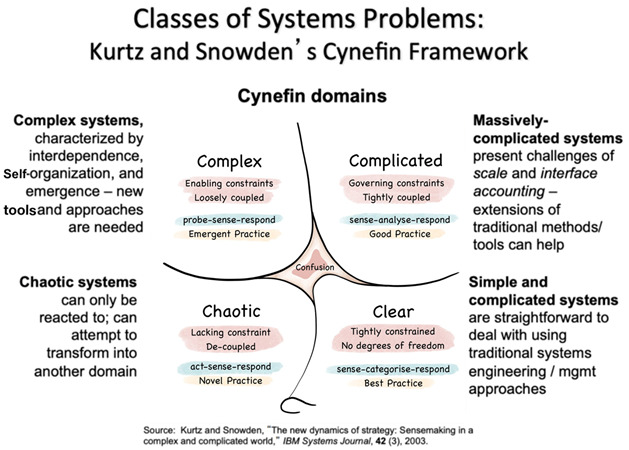

Understanding the complexity of a target system, or subsystem, can help in determining how to analyse, or how to act. Focusing on the quadrants in the schematic above, traditional behavioural science works in the bottom right domain—‘known” (or simple). This is where the situation is relatively simple and one can be confident of the cause and effect relationships holding true. As such, we can develop habits, redesign the physical environment, and use other intervention tools to change behaviour systematically. Moving up to the top right domain (and to some extent the top left domain also), ‘knowable’ and ‘complex’ systems are where we might apply our systems thinking approaches. There are more factors at play that can influence behavior, and so any one change or intervention might have unpredictable knock-on effects and potentially cause unintended consequences (or no consequences). This is where we need to understand some extra concepts and elements in order to design and implement changes. The next section will introduce those. Finally, for completeness, in chaotic situations, perhaps reflecting an acute shock or crisis (bottom left), systems thinking is of no use and other approaches, beyond the scope of this document, are required. The very early days of the COVID-19 pandemic as countries closed their borders and introduced population restrictions can be considered a somewhat chaotic system from this perspective.

### 4.1. Boundaries and a Holistic Perspective

Recently, the World Health Organisation published a guide to approaching non-communicable diseases through systems thinking, introducing different systems modelling tools, and acknowledging that complexity exists on multiple scales and frames. One example, depicted in Figure 1 and explored further below, offers different but complementary ways of viewing a system ([34]; [68]).

As reflected in Figure 1, boundaries are critical for systems thinkers. For example, boundary judgements are required in order to identify the scope of the system to be studied, whether it contains subsystems, and how these inter-relate. Constraining a system can be useful as it (artificially) reduces the complexity whilst still allowing the exploration of the relationships and interactions between its components, emphasising the inter-connectedness of elements. Zooming in and out of the granularity of a system or subsystem provides insight, as does the recognition of hierarchies within a system. However, the boundaries of complex systems can be fluid, changing over time. This can make it challenging to define and manage the system’s scope. The bigger the system and the more elements, the greater the complexity of the system. Even though boundaries can be fluid, any approach requires clear identification of the boundaries of the system for the purpose of the intervention or analysis. These judgements should be made by actors with a diversity of worldviews relating to the system in question, as well as a consideration of who can affect change, and who is affected by change. These constraints can be enabling, i.e., they give clear indication as to what is in and out of scope.

For example, the Foresight obesity map 2017 (below Figure 2; https://www.gov.uk/government/publications/reducing-obesity-obesity-system-map accessed on 31 January 2025) is huge, and in many ways is too unconstrained to be of value to an interventionist. Though it is excellent as a map showing key elements and actors influencing different parts of the overarching system, in order to work more effectively, a practitioner might zoom in to a specific loop in the system. For example, focusing on physical activity (in orange, to the right of the central ‘energy balance’) might provide a more tractable ‘constrained’ system to work within on an applied project. This reduces the complexity by reducing the number of actors and processes requiring attention. In this way, systems thinking emphasises the importance of defining clear boundaries for the system under study, but at the same time recognises the importance of the supra-system, or larger open system where for example, unintended consequences may occur.

### 4.2. Interconnections, Interdependencies and Emergent Properties

Systems thinking recognises that elements within a system are interconnected and interdependent. Changes in one can have ripple effects on others and thence overall system behaviour. This reflects the shift from simpler systems to more complex systems (perhaps changing the behaviour of one individual in a constrained situation compared to changing the behaviour of a team within the context of a larger workplace culture), though as humans themselves are complex open systems, there is a certain level of fractality when zooming in and out. This has been recognised and developed by as is recognised by Beer in his Viable Systems model through the inclusion of a recursive structure ([2]). Understanding the relationships and dependencies between components is crucial for analysing how changes propagate through the system and for identifying leverage points for intervention or improvement ([41]; [45]).

As the above infers, systems can exhibit emergent properties, which are properties or behaviours that arise from the interactions of system components but are not directly attributable to any individual component. These emergent properties cannot be understood by analysing the individual parts in isolation and require a holistic perspective to uncover the underlying dynamics. This also means that systems are self-organising and adaptive, which simply means that when elements of a system change, the larger whole will find a new equilibrium without any form of management or intervention. This is why classical approaches to changing a system often have little noticeable impact—the system itself simply ‘absorbs’ the intervention changes. Some approaches and tools can be particularly effective here such as zooming in and out, as well as probing the system and sensing the impacts, and then responding with an intervention.

### 4.3. Feedback Loops

Feedback loops play a crucial role in systems thinking. They represent the circular causal chains relating to information flows within a system. Feedback loops can be reinforcing (positive feedback) or balancing (negative feedback). Positive feedback loops amplify change, while negative feedback loops help maintain stability or balance within the system. Analysing feedback loops helps in understanding system behaviour and identifying opportunities for intervention. For example, one might want to reinforce positive feedback loops if they are of value in increasing the target behaviour, or disrupt them if they are not; how one intervenes depends on the desired outcomes. One additional layer of complexity is that there are often time delays, or lags, embedded within feedback loops, which means it becomes harder to predict and manage behaviour influenced by the loop.

System mapping and causal loop diagrams are graphical tools used in systems thinking to represent the causal relationships and feedback loops within a system (see the obesity map above), and are part of an array of methods within a system, thinking toolbox ([68]). These diagrams visually depict the cause-and-effect relationships between system components and help in understanding the dynamics and behaviour of complex systems. Causal loop diagrams provide a means to communicate and analyse the structure of a system and identify key elements and their interactions.

### 4.4. System Equilibrium

Generally, all dynamic systems seek a state of equilibrium ([3]). However, due to the complex ‘moving parts’, there is always tension within the system, and often behaviours are triggered by a need to re-establish a steady state. Actors, behaviours, and system rules all create energy, and whilst we usually wish to maintain an optimum state of equilibrium, the tensions that arise can also help identify leverage points where system change can be accelerated. An analogy can be drawn with the membrane potential at a synapse which maintains an equilibrium until a certain energy potential is reached. A neuron will only ‘fire’ once there is sufficient energy generated from its inputs that brings the membrane potential to its specific threshold. Once this threshold is reached, the whole ‘system’ of the neuron changes and an action potential is triggered. In the same way, a system will remain relatively stable even with a certain level of tension and energy within it. However, once a threshold is reached, an ‘action potential’ is triggered and the system changes. These tipping points reflect a critical element of change within a complex system ([44]).

### 4.5. Requirements for System Change

[46] ([46]) describes several ways to shift a system to a new state. These are primarily focused on the system parameters themselves (including the rules of the system) and less focused on directly targeting the behaviours of actors within the system. Nevertheless, it is useful to consider some of these high-level approaches. Firstly, the most profound leverage point is in the mindset or paradigm out of which the system arises. Changing the way people think about and perceive the system can lead to transformative change in actors within the system. Similarly, adjusting either the rules or the goals of the system, or what the system is striving to achieve, can significantly impact its behaviour and outcomes. This latter can be equated to the higher rungs of the Nuffield intervention ladder, e.g., regulation and policy changes. However, these two high-level approaches are likely impossible to realise in the real world as they require a fundamental change in the rules or structure of the system (though see below with reference to the Nuffield intervention ladder as well as Chater and Loewenstein’s s-frame interventions ([52]; [7])). Improving information flows within the system can enhance transparency and feedback, leading to more informed decision-making and better outcomes. And likewise, strengthening or weakening feedback loops can help stabilise or destabilise the system to move towards a desired goal. Reinforcing positive feedback can amplify desired changes, while balancing feedback can mitigate unwanted changes. reducing delays in the system’s feedback and response times can help it react more swiftly and appropriately to changes.

Critically, cause and effect may only be observable in hindsight. The system will change over time and so associations that held true previously may no longer do so. Uncertainty and ambiguity are likely. A good-practice approach in these situations is to value *progress* towards a goal, rather than reaching the endpoint in itself. Furthermore, systems tend toward equilibrium—though this may not be the one desired—hence the need for system change. Attempts to move the system into a new state sometimes fail because energy is not focused on the point in the system with the greatest potential for change. This is sometimes called a tipping-point, or a key leverage point ([46]). If everyone within the system shares a narrative of stability, then change will be an uphill struggle. However, if there are noticeable pockets of dis-equilibrium where tension exists—either demonstrated through conflicting narratives, or by clear structural conflicts—then intervention and pressure at this point is most likely to result in change. [1] ([1]), provide an example from the domain of sustainability—they note that most interventions that attempt to change group behaviour focus on seemingly easy targets but that are in actual fact weak leverage points, and so have little impact. Instead, they argue that any attempt to transform a system’s dynamics should first consider and identify where the strongest leverage points reside. These can be structural, e.g., changing the rules, the system goal, or the system values/mindset ([45], [46]) or they can be dynamic, e.g., where there is disequilibrium and tension already present in the system. For example, some have argued that the Arab Spring was enabled by momentum produced through social media that leveraged a point of tension in the system ([36]). A key lesson here is not to fixate on where the problem appears to arise but seek to understand what the factors are that maintain the equilibrium. Additionally, to consider how and where to exert leverage to create a tipping point—where in the system might intervention take place, leading to a significant switch from one state to another.

Ways in which a human-centred approach can integrate with the above lean heavily on the significance of behaviour in maintaining the equilibrium within a system. Perspective is an important lens and seeing the whole system (as determined by boundary judgements), rather than parts in isolation, enables a change agent to zoom in and out of the system to consider the detail at a granular level, but then also zoom out to appreciate the additional interdependencies and interactions at a higher level. This allows a probe-sense approach to understanding the unintended consequences as a result of a targeted nudge. Likewise, acknowledging that causality may not be linear or simple, and hence looking for (emergent) patterns across the system and over time is productive. Again, using a systems thinking tool such as visual modelling (system maps, causal loop diagrams, etc.) that emphasise the actor’s behaviour within a system is valuable.

In general terms, human behaviour is best maintained when there is immediate feedback—falling off a bike and grazing one’s knee is a powerful piece of feedback to aid quickly learning to ride a bike. Because many systems have delayed or slow feedback loops, perhaps due to processes and rules within the organisation, undesired behaviours may proliferate. Speeding up feedback is one way to enhance system change to desired behaviour. As mentioned above, designing interventions that operate at multiple levels of the system, including individual, organisational, and policy levels will also likely have a greater impact.

To facilitate of sustained change, based on work in design thinking and implementation science, we must ensure any intervention is human-centred and co-produced with actors in the system ([57]). Engaging stakeholders, including community members and end-users, in the design and implementation of interventions ensures that interventions are contextually relevant and more likely to be adopted and sustained. This process, if authentic, can build trust, consensus and a shared narrative for change (Box 2).

Box 2Some system leadership characteristics.Reflect regularly:Plan, Do, Study, Act ([14]). Working within a complex system needs constant consideration of the changes in dynamics. This is particularly true once a system change intervention has begun. Continue to learn and be curious about the system. Reflect and refine for ongoing improvement.Build consensus and Trust:Systems change when thresholds are exceeded, and tipping points reached. By engaging partners and stakeholders across the system, consensus and common action builds momentum. Single points of pressure are less likely to work. Implement robust governance structures and share values.Develop a shared understanding and consistent language:Develop a shared understanding of the system and local context. Individuals from different disciplines or experiences will have different terms and associations for different concepts. Use a consistent language to reduce ambiguity and confusion. Use boundary objects and share narratives and perspectives. Have meaningful engagement with local communities who experience the system to understand their viewpoint.Understand the power of narratives:Understand the power of personal narratives. Listen to what people say about the system: what are the key dynamics, what are the recurrent themes? Narratives act as cognitive schema or habits that maintain a personal sense of equilibrium for individuals within the system. Change the narrative and change the system.

## 5. Methodological Approaches

### 5.1. Analysis and Experience

Methodological approaches to behavioural science and to systems thinking tend to differ as one has historically focused on the individual, whilst the other on a much broader piece, including and extending beyond the human sphere. Nevertheless, there is overlap and clear scope for integration. Accepted methods, or ways of working, within each discipline can be split between analytic (gathering data) and interventional (designing and implementing an intervention for change). The latter also encompasses elements of implementation science ([57]). Additionally, a potential overarching framework would be Demming’s Plan-do-check-act (PDCA) approach ([14]) which is a form of action research, or a sense-probe approach allowing for navigation in complex waters. This approach which is currently popular within health domains as ‘total quality management, or improvement’ ([6]) allows for a more fluid and agile approach to complex challenges, much like PDCA has been argued to provide. That said, traditional approaches to data gathering in the behavioural sciences tend to involve surveys, focus-groups, and interviews, as well as the use of observational data to validate actual behaviours. Such data can then be analysed through various behavioural lenses such as the COM-B framework ([47]), yielding insights into barriers and enablers to change, as well as particular thematic avenues for intervention (e.g., motivation, capability, etc.).

Analytic methods in systems thinking attempt to gather data from across different elements of the system, often with an emphasis on a different aspect of the system structure. Often, the goal is to create a ‘systems map’ that identifies relationships within the system between actors, processes, structures, and policies. For example, Causal Loop Diagrams are visual tools that depict feedback loops within a system, illustrating the cause-and-effect relationships among variables. CLDs are instrumental in identifying reinforcing (positive) and balancing (negative) loops that drive system behaviour ([60]; [61]). At a glance, it can be seen that the behavioural toolkit (surveys, observation, etc.) can be integrated with systems thinking methods (e.g., CLD) to provide behaviourally informed systems maps and the like. Other systems thinking approaches include behaviour over time (BOT) graphs which plot a variety of chosen variables against time, visualising trends and patterns in system behaviour (human and process). BOT graphs help identify long-term trends, cycles, and potential tipping points ([39]). Likewise, Leverage Points Analysis identifies places within a system where small changes can lead to significant impacts. These leverage points are crucial for effective intervention and system redesign ([45]). Maps can be simple and qualitative, perhaps based on stakeholder discussion and feedback, or can be quite complex and computational. For example, in the former camp is the ‘rich pictures’ approach. These are informal drawings that capture the qualitative aspects of a system, including relationships, conflicts, and interactions. They provide a holistic view of the system and engage stakeholders in the analysis process ([8]; [48]). Towards the more computational end, Agent-Based Modelling simulates the interactions of individual agents within a system to understand how their behaviours affect the system as a whole. ABM is useful for studying complex adaptive systems, where agents follow simple rules but produce emergent behaviour ([21]). Alternatively, Behavioural Systems Mapping (BSM) identifies actors, behaviours, feedback loops, and system rules within complex systems. BSM is an effective step towards integrating systems thinking and behavioural science in its methodology ([29]). For further details on these approaches and how they can be integrated, the WHO report on NCDs provides a good introduction ([68]) and the Busara project provides a real-world worked example ([18]).

### 5.2. Intervention

The acknowledged gold standard intervention in the human sciences tends to be the randomised-control trial (RCT), which provides rigour, a robust set of methodologies, and reliably high-quality data. However, it can be difficult to apply a pure RCT approach to a system due to its complexity and likely unique nature. Not only is it challenging to identify a control condition, but it is also near impossible to create ‘blind’ conditions, and the system will likely change over time. Furthermore, it is argued that linear cause and effect breaks down as complexity increase, making it difficult to draw conclusions. Nor will lessons from one system necessarily generalise to other systems. This has led to Schmidt and Senger ([58]) arguing that traditional RCTs are too ‘brittle’ for use in complex systems, in that they may not transfer contexts, they may not find equilibrium within the larger system, and they may only have a short-term impact as system dynamics change. As such, there have been proposals of how to adapt RCTs is such situations. For example, [11] ([11]) developed a Sequential Multiple Assignment Randomised Trial (SMART) in order to evaluate more complex health interventions where the system may vary over time. Likewise, Hallsworth, has argued for an evolution of RCTs to acknowledge the complexity in systems ([31]). As such, an alternative approach, acknowledging the truly complex dynamics within a system, is to accept that quantifiable causal relations will be difficult to identify (and may not last), and instead rely on other forms of data emanating from the system. For example, Snowden ([66]) has argued that narratives within the system give a powerful insight into the dynamics and relationships, as well as identifying where tension exists and key tipping points which may be leveraged. Perhaps the ideal practice is to combine quantitative measurements of behaviour within the system in a systematic way, along with rich qualitative data providing additional layers to understand system dynamics. Ultimately, a successful intervention methodology wishes to gather (1) behavioural evidence of changes to the desired behaviour being carried out within the system, and (2) evidence of the system finding a new equilibrium.

Finally, to successfully navigate a complex adaptive system, an emphasis can be placed on certain characteristics within the broader intervention methodology design. These include iterative testing and adaptation, i.e., implementing interventions in a phased manner, using pilot programmes and small-scale tests to gather data and refine approaches. This iterative process allows for continuous learning and adaptation based on what works and what does not. Also, cross-disciplinary collaboration, through fostering collaboration between experts in behavioural science, systems thinking, and other relevant fields. Interdisciplinary teams can bring diverse perspectives and expertise to the design and implementation of change initiatives. In a complex space, it is necessary to probe and sense ([65], [66]) to identify change. As such, robust methods for measuring the impact of interventions on both behaviour and system outcomes at small and larger scales are valuable. Using data to evaluate effectiveness, identify areas for improvement, and demonstrate the value of integrated approaches is important before iterating the next phase of the intervention. Finally, change requires trust, consensus, and shared goals. Therefore, building capacity and awareness becomes key. Educating and training stakeholders, including policymakers, practitioners, and community leaders, in both behavioural science and systems thinking is an important meta-goal. Building capacity and awareness helps create a shared understanding and commitment to integrated approaches.

## 6. Role of Design Thinking

There are previous examples of how the Design Council’s double diamond (https://www.designcouncil.org.uk/our-resources/framework-for-innovation/ retrieved 11 June 2024) approach to design thinking has been used to help develop behavioural interventions ([12]; [53]; [62]; [67]). It is a methodology designed to be used on complex systems and aligns well with various elements of systems thinking, e.g., perspectives, consensus building, narratives, etc., (see Figure 3). A critical strength is that design thinking is a user-centred approach and often incorporates co-design and co-production of solutions with key stakeholders. It also adopts a sense-probe approach, aligned with operating within complex systems. Finally, it is iterative and provides an agile way to adapt to dynamic changes over time that can occur in some systems.

Goodman and colleagues have developed it as a core component within applied behaviour change approaches, as well as when working with new technologies ([53]). It has also been used to understand patients journeys during acute care in hospitals, emphasising the patients journey from a user-centred perspective ([62]). In both these cases, a novel multidisciplinary approach was adopted to harness the strengths of multiple perspectives, user-centred co-design, and iterative intervention development.

More broadly, design thinking and systems thinking are complementary approaches that can significantly enhance the development of effective interventions ([67]). For example, part of the user-centred focus surrounds empathy, and the development of a deep understanding of the needs, experiences, and emotions of the people operating within the system. Additionally, research biases and prejudices are overcome by emphasising the experiences of the actors within the system. Likewise, involvement of users during ideation and design phases increases the likelihood of adoption by the end user. Ideation encourages the generation of a wide range of ideas and potential solutions. In systems thinking, ideation can help identify innovative leverage points and strategies for intervention. Prototyping involves creating tangible representations of ideas to explore and test their feasibility, and can overcome challenges in the lack of clarity around cause and effect relationships, as well as the complex nature and pattern of system dynamics. In systems thinking, prototyping can help model parts of the system or specific interventions, allowing for experimentation and refinement. As noted, intervention testing gathers feedback on prototypes from stakeholders and users. This phase is critical in systems thinking as it provides insights into the effectiveness and impact of the intervention, allowing for adjustments and improvements. Finally, both design thinking and systems thinking emphasise an iterative approach. Continuous feedback and refinement cycles ensure that interventions remain responsive to the evolving dynamics of the system and the needs of its stakeholders. Such collaboration leverages the diverse perspectives and expertise of different stakeholders. In systems thinking, collaborative efforts can lead to more holistic and inclusive solutions that account for various viewpoints and interests.

## 7. Intervention Scale and Scope

What is the nature of the change we seek? And how might we answer this question? For a behavioural scientist working within a constrained boundary (perhaps a workplace, or a specific target group of citizens) a system will usually be relatively stable and have found its own equilibrium. Even if we seek to make a smaller change with the system, we might actually have the (un)intended consequence of shifting the whole system. Nevertheless, our problem may be specific and closely bounded to a specific context or situation and so we might seek to make a change within a system whilst leaving the overall situation intact. Alternatively, if the system we are studying is clearly ineffective or poorly designed, delivering outcomes that are counter to those desired, then our ambition may be broader. Either way, we might use similar tools and analyses because the changes we desire will likely be either behavioural (of the people acting within the system) or structural (the rules of the system itself). The key here in identifying appropriate behavioural measures is to evidence change followed by designing targeted interventions across layers of the system to achieve the desired change.

Within the ‘nudge’ literature, there is an ongoing discussion about the level at which to intervene in changing the behaviour of an individual. For example, at one extreme, should one introduce regulations to prevent a behaviour from occurring (e.g., the banning of transfats in food production so that people cannot eat it). Or at the other extreme, provide information which can inform individual choice but not control it (raising awareness that saturated fats are damaging to health). These considerations were codified by the Nuffield Council on Bioethics in 2007 with the introduction of their ‘intervention ladder’ (see Figure 4) ([52]).

In her book, *Thinking in Systems*, Meadows argues that the most powerful ways to change a system are through changing the rules or the game ([46]). Reflected here, the top levels of the ladder would equate to a change in the system dynamics. Within the health domain of food choice, overweight and obesity, the introduction of sugar taxes, or banning certain foods altogether (e.g., sugar beverages, saturated fats, etc.), would fundamentally change the overarching complex adaptive system. Currently, Western governments tend to step onto the lower rungs when considering preventative health behaviours of their citizens and push responsibility onto individuals to ‘make the right choice’.

In their paper “The i-frame and the s-frame: How focusing on individual-level solutions has led behavioral public policy astray” ([7]), Chater and Loewenstein argue that expecting individuals to bear the burden of responsibility for system change detracts from a broader questions around system efficacy and design. It is difficult to behave in a particular way if the system is designed to drive behaviour in a contrary direction. Following this line of reasoning, we cannot expect individuals to change their eating behaviour when the system incentivises production and consumption of unhealthy foods (farm subsidies, lack of regulation in food production, etc.). This necessary recognition of the role of system rules serves to temper our expectations of intervention impact and also helps frame where responsibility lies for different elements of change. It also helps avoid the danger of focusing energy on weak leverage points that appear easy targets ([1]). Nonetheless, analysing and identifying potential interventions provides a richer understanding of the overall system and its dynamics. For example, we might identify one solution to unhealthy eating as being the regulation of sales of obesogenic foods. However, recognition of the intransigence of politicians and the power of industry lobby groups might lead to the conclusion that energy would be wasted in focusing on this type of solution (a poor leverage point). When co-designing interventions, it is important to build in a consideration of these system dynamics and to explicitly test whether the rules-of-the-game are open for change, or whether an intervention will necessarily need to be more organic. This provides a reminder of a couple of key concepts of systems thinking, namely finding leverage points where changes in momentum are more likely to produce results, and building bottom-up momentum through shared narratives, trust building and identity. There is some laboratory evidence that the adoption of a new social habit or norm only needs around 30% of a population before it becomes pervasive ([4]), though the precise proportion needed likely depends on the context, particularly whether the new behaviour happens to target a point of tension and hence a potential tipping point. A behaviourally informed system intervention may therefore be best designed to target a tipping-point (often defined as situations where the system narrative is unstable or ambiguous) and building grass-roots momentum through a campaign of trusting building—a ‘coalition of the willing’—in order to generate sufficient momentum for change.

## 8. Real-World Examples

### 8.1. Diabetes System Mapping in Wales

One contemporary example of using behavioural systems maps (BSM) focuses on diabetes care in Wales. A partnership between Public Health Wales, The Cwm Taf Morganwg University Health Board (CTMUHB) and University College London (UCL) explored the complex interplay of actors, behaviours, and processes within the broader health system looking at treatment and care for individuals diagnosed with type 2 diabetes. The system comprised 25 different actors incorporating around 150 behaviours, along with behavioural influencers such as social norms and workplace culture, facilities and equipment, and formal processes within the healthcare system. The mapping exercise surfaced key interdependencies as well as identifying barriers within the system. Of the latter, some processes appeared to be siloed with a lack of a whole-system approach from healthcare professionals. Duplication across processes was also identified as was a lack of communication in some areas where important interdependencies existed, such as in key learning ‘teachable moments’ between professionals and patients. This final item might be considered a leverage point where a small intervention could make a significant change to the effective functioning of the system. Behavioural recommendations included relevant training and also identifying opportunities for the sharing of good practices including networking and relationship building.

### 8.2. Decarbonising Housing Stock

A second example focuses on decarbonising the housing stock in Wales ([29]). In recognition of the need to reduce carbon emissions, along with the acknowledgment that poorly insulated and inefficiently heated homes was a key source of emissions, BSMs were created for private rental and owner-occupied houses. The system of focus was on retrofitting existing houses—this provides a clear constraint to the system in question and reduces the complexity. It also helps to identify actors and behaviours of key relevance. Causal loop diagrams helped to identify feedback loops and key points in the system for proposed intervention. Additionally, the COM-B framework ([47]) was used as a behavioural lens to interpret the maps, particularly in framing outputs. For rented accommodation, the landlord, as an actor, was identified as a key leverage point for positive feedback, with much influence coming from systems rules (e.g., whether there are incentives in place to support the costs of retrofit). In contrast, owners were important actors for owned homes, and influences on them included factors such as social norms and peer influence. A series of workshops, building from problem specification to proposed solution, resulted in 10 policy recommendations that spoke to elements of capability, opportunity and motivation.

These two examples provide insight into how concepts in systems thinking (interdependencies, causal loops, actors, etc.) and behavioural science (motivation, capability, social norms, etc.) can work together in both understanding the problem in a complex system but also in identifying potential solutions, or intervention points.

## 9. Synthesis: Behaviourally Informed Systems Thinking

### 9.1. An Integrative Framework

Through the integration of systems thinking and behavioural science, using design thinking as a framework, a route map with defined checkpoints can be developed (Figure 5).

What becomes clear from considering systems thinking and behavioural science side-by-side is that many of the key behavioural drivers acknowledged by contemporary models, such as COM-B, play a prominent role in understanding systems. For example, the role of environmental triggers, social identity and norms, and existing habits or routines reflect elements of a complex system that maintain its equilibrium. At one level then, integrating behavioural science is not a qualitatively new approach, but one of emphasis and selection. Design thinking brings an overarching framework to the methods, as well as emphasising the importance of a user-centred approach. In this case, ensuring an understanding and the involvement of the actors who operate within the system.

COM-B is an effective lens through which to view the drivers of behaviour, exploring the barriers and enablers that can be used to shape the behaviour of an individual, or target group, effectively. It can also be used at different levels of a system to understand the broader dynamics and interdependencies. Looking at the ‘problem’ from different perspectives, and by zooming in and out, COM-B can identify underlying drivers at different levels. It can be used with system maps to understand interdependencies, relationships and patterns, and can be used, for example, within focus groups, to thematically categorise narratives and common experiences. Other behavioural insight tools can be integrated. For example, an understanding of social norms within the system can help uncover who the key and influential messengers are, where there are leverage points within the system narratives, and where and how consensus might be built. Likewise, a consideration of ‘defaults’ can identify how a system equilibrium is being maintained, as well as identifying leverage points for change. Relatedly, the habits of individuals within the system can interact in unexpected ways and create emergent properties. Critically, these approaches and tools need to be applied across levels of the system (zooming in and out) in order to obtain a better understanding of the system dynamics.

One of the most important messages from dual-process theory is that much of our behaviour is driven by unconscious triggers ([37]). For a system, this means that the rules and structure in place will dictate a large proportion of behaviour, irrespective of the intentionality of the individuals involved ([27]). As such, dual-process theory is foundational in understanding how to influence human behaviours in a complex system such as designing the environment within the system to unconsciously produce the desired behaviour. We can consider a high-level approach to designing elements within a system based on how and in what circumstances we want to shift behaviour (Figure 6) ([20]; [27]). This approach can then be applied at leverage points, or alternatively in areas where greater stability is required.

### 9.2. Discover, Define, Develop, Deliver

In the early stages of the process, data will be gathered through surveys, interviews, observation, and through mapping system structure and properties. Some methods can be taken from traditional systems thinking approaches (e.g., identifying core narratives maintaining equilibrium, causal loops diagrams and other system mapping), whilst others will be behavioural (e.g., a consideration of the contributions of Type 1 and Type 2 processes in system behaviours, analysis of capabilities, opportunities and motivations in system actors). This combination generates a richer dataset to gain insight into the specific characteristics of the system that are maintaining an equilibrium with target behaviours. With system thinking expertise involved, it is possible to build functional models, as well as computational models, in order to explore system behaviour when parameters are changed. This can be particularly useful in identifying leverage points and understanding key actors or processes or structures within the system ([45]).

At this point, the researcher can move from a high-level statement about what is wrong with the system, to a more precise statement of what problem the intervention is going to tackle. For example, moving from ‘there is too much obesity in this organisation’ to ‘people choose unhealthy foods in the canteen for lunch given the choice they have and the time available’. The latter then allows for the creation of a precise problem statement, given in behavioural terms, such that targeted interventions can subsequently be designed to address the issue. From the design thinking perspective, this allows the mapping of a problem-solution pair.

Solutions require creativity and ideation; if intervention design is based solely on prior assumptions, then it will likely lead to a bad behavioural intervention. A powerful approach to generating ideas is by asking “How might we…?” questions. Such open questions necessarily lead to the generation of multiple possible ideas ([40]). As such, the next stage of the process is to use the data gathered in the first phase, as well as revisiting sources, such as stakeholders, the system maps, and behavioural analyses, in order to identify potential interventions. These can then be evaluated, ranked, combined, and re-iterated in order to identify a design for testing. By involving stakeholders and their narratives, it is possible to unearth places of tension within the system and hence potential leverage points. Likewise, new narratives can be explored, as can areas that might develop into a consensus and energy related to change. The behavioural toolkit can be unpacked to identify Type 1 and Type 2 interventions that will help generate or catalyse energy at leverage points. For example, the use of social norms in behavioural nudges is a powerful way to influence the behaviour of larger groups of individuals within a system. Likewise, changes in physical architecture or in defaults can shift the routine behaviours being triggered in specific locations and contexts. As such, combining insights from these two broad approaches can enhance the efficacy of co-designed interventions. An intervention, which may be made up of an array of system shocks or nudges, can then be introduced to the system. In complex contexts, a probe-sense approach is encouraged ([41]), which translates into an iterative approach to change. Indeed, the standard RCT approach to measuring change may not be appropriate in a dynamic system, as clear linear causality breaks down ([58]). Instead, a more dynamic, qualitative and agile approach is needed, and may include agile action research approaches, further gathering of narratives as well as other qualitative measurements, and also direct observations of whether actor behaviours change as desired.

Interventions should ideally be co-designed with actors from within the system and a segmented approach may prove valuable where minority groups hold conflicting narratives from the mainstream. Stacking or combining types of intervention may be more effective, particularly if applied across different levels of the system ([43]; [64]), and approaches to culture change emphasise the human-centred nature of effective system shifts ([13]). (See Box 3).

Box 3Human-centred behaviour change.Organic, or bottom-up, system change requires a coalition of similarly minded individuals with sufficient energy to build a momentum towards to tipping point. Meadows ([46]) emphasises the need for changes in mindset across the system. As such, the development of a new shared narrative of the required change along with building trust amongst the group is critical. Snowden ([66]) presents the idea of generating ‘more stories like this’ to propagate a narrative for change across a system in order to generate energy and reach a tipping point. Likewise, identifying role-models and messengers who have the greatest impact on actors within different parts of the system will help to propagate the narrative. The use of an analytic tool (such as COM-B or a systems map, or both) to identify key leverage points within the system will ensure effort is not wasted ([1]). A parallel consideration of the need for structural change is also worthwhile (either at a high level: Do the rules need to change? Or at an operational level, are there architectural/ contextual features that are triggering the wrong behaviours?). We must introduce an intervention at a leverage point and measure the response. Then, we must consider unintended consequences. Finally, we must amplify positive effects through identified feedback loops, and the use of classic behavioural tools (social norms, messenger, habits, incentives, salience, ego, etc.) ([20]).

Implementation requires the consideration of various criteria such as the acceptability of the intervention within the system ([35]) as well as a review following roll-out. Further iterations may be required in order to consider factors such as unintended consequences, or areas where the system map proves inaccurate in its modelling.

## 10. Conclusions

There is a clear case for integrating systems thinking and behavioural science: they share overlapping constructs, some ideological perspectives, and even employ some of the same approaches and tools. Whereas behavioural science has traditionally focused on the individual, systems thinking focuses on the many. Many of society’s ills are complex in nature (health, access to resources, sustainability, equality) and involve many individuals across cultures and life situations. Both are multidisciplinary by nature and a combination of the two, when applied appropriately, provide a complementary approach to understanding and solving complex problems. This review has identified high level concepts and structures to both approaches and presented ways in which the two can be integrated. It has also identified areas of differentiation and divergence. Finally, it has highlighted design thinking as a practical framework and process to help integrate the two approaches and to structure problem-solution mapping ([53]).

Historically, systems thinking and behavioural science have been considered to require different mindsets or ideologies, such as with regard to methods, analysis, and scaling ([25]; [49]). Indeed, some have argued that behavioural science is seen as a ‘last resort’ when all else has failed ([25]). It is certainly clear that there are some ideological challenges to overcome in combining the two approaches. For example, Meadows and others have argued that ([45], [46]) the best leverage points for a system are by changing the rules of that system, or influencing other high-level elements such as the mindsets of system leaders. Alternatively, Snowden has argued that leverage can be gained and tipping points exploited where social and behavioural tensions exist within the system itself ([41]; [66]). The latter represents a bottom-up approach to system change that is more akin to behavioural science interventions such as through social norms, role modelling or framing. These views align with systematic and systemic (hard vs. soft) approaches to systems thinking ([26]; [34]; [56]). In behavioural terms, whilst governments might regulate a system using a ‘hard’ engineered approach (e.g., introducing a sugar tax on processed foods), most behavioural scientists will be using a soft systems approach to identifying leverage points to intervene in order to bring about change.

Another key area of difference is found in methods for intervention and evaluation. The gold standard in behavioural science is the randomised control trial, which aims to reduce confounding factors in determining the significance of a defined intervention. In contrast, such an approach is considered inappropriate in systems thinking as systematic causality can break down as complexity increases. In such cases, a more dynamic approach is required ([65]), for example including action research, looking for shifts in overall system behaviour, or extracting narratives from key actors within the system. Ultimately, these differences are driving innovation in methods, such as the SMART methodology for delivering an RCT-like evaluation in a complex dynamic system ([11]), or the use of actor narratives to identify leverage points for intervention ([66]).

Nonetheless, there are many points of contact between the two approaches, including the actors, behaviours, feedback loops and rules that together form the system under observation, and the behaviour that is exhibited within these operating conditions ([46]). Integrating behavioural science with systems thinking is essential for addressing the complex, interconnected challenges—often referred to as “wicked problems”—that our societies face today. Behavioural science offers insights into the drivers of human actions, while systems thinking provides a framework to understand the dynamic and interrelated nature of complex societal issues. For example, together they provide a better understanding of how cognitive biases can produce large-scale behavioural effects such as through social norms ([55]), or indeed may shed light on political decision-making which might seem irrational to a dispassionate analysis (e.g., regulation on classified drugs that does not map onto scientific evidence of harms and benefits) ([50]; [51]). Finally, the integration of behavioural science and systems approaches allows for more comprehensive and adaptive solutions by considering individual and group behaviours within the broader context of societal, environmental, and policy systems. This integration not only deepens our understanding of the drivers of complex problems but also enhances our ability to design solution-focused interventions that are more effective, sustainable, and equitable.

## Figures and Tables

**Figure 1 behavsci-15-00403-f001:**
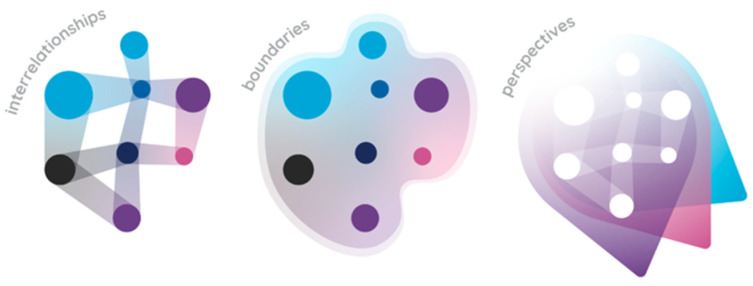
WHO non-communicable disease report: multiple layers of complexity. [This work is available under the Creative Commons Attribution-NonCommercial-ShareAlike 3.0 IGO licence (CC BY-NC-SA 3.0 IGO; https://creativecommons.org/licenses/by-nc-sa/3.0/igo accessed on 31 January 2025)].

**Figure 2 behavsci-15-00403-f002:**
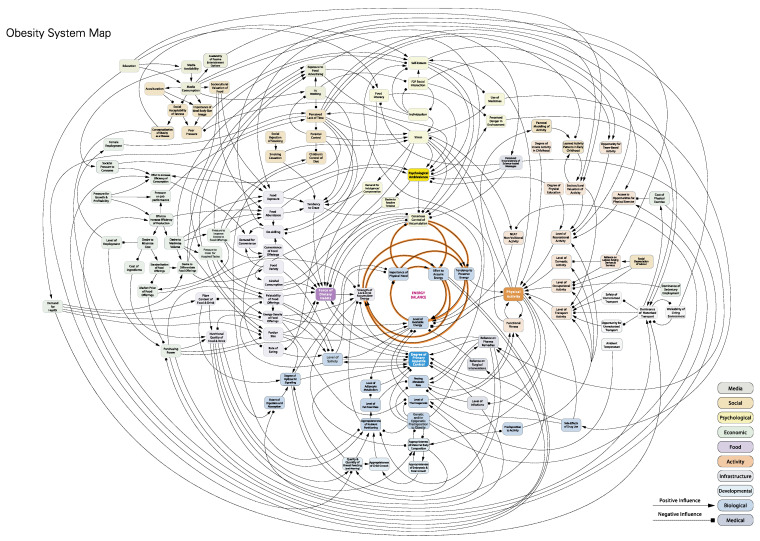
Foresight Obesity Map 2017.

**Figure 3 behavsci-15-00403-f003:**
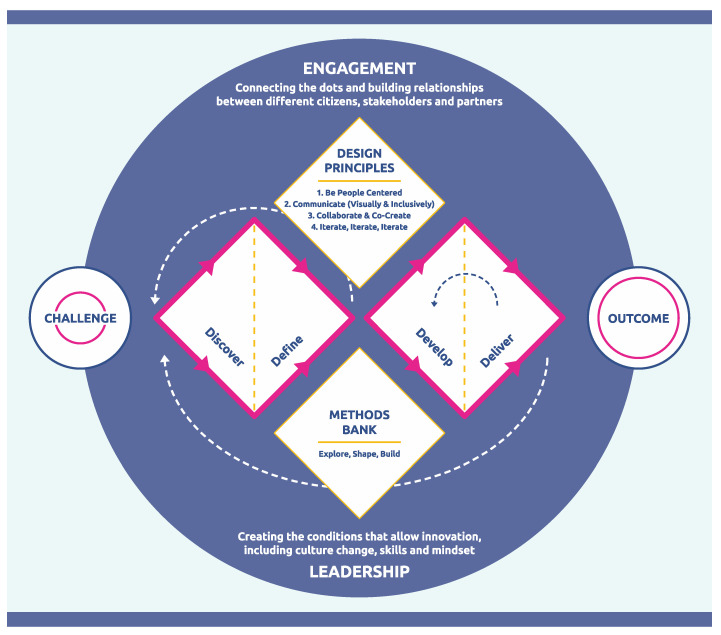
Adapted from https://www.designcouncil.org.uk/our-resources/framework-for-innovation/ Creative commons licence (CC BY 4.0 license accessed on 31 January 2025).

**Figure 4 behavsci-15-00403-f004:**
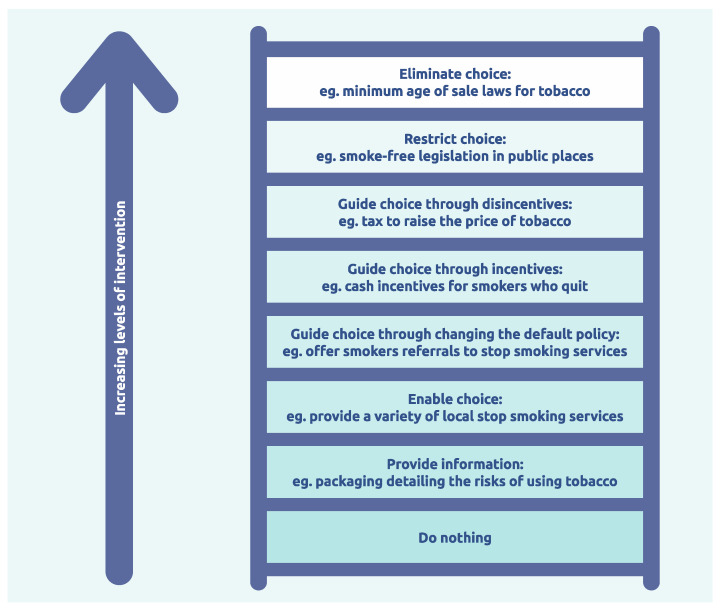
How forcefully to intervene. Adapted from “What role do taxes and regulation play in promoting better health? Licenced by The King’s Fund under a Creative Commons Attribution-NonCommercial-NoDerivs 4.0 and accessed on 31 January 2025. https://assets.kingsfund.org.uk/f/256914/x/5016d2a134/what_role_do_taxes_regulation_play_better_health_2020.pdf.

**Figure 5 behavsci-15-00403-f005:**
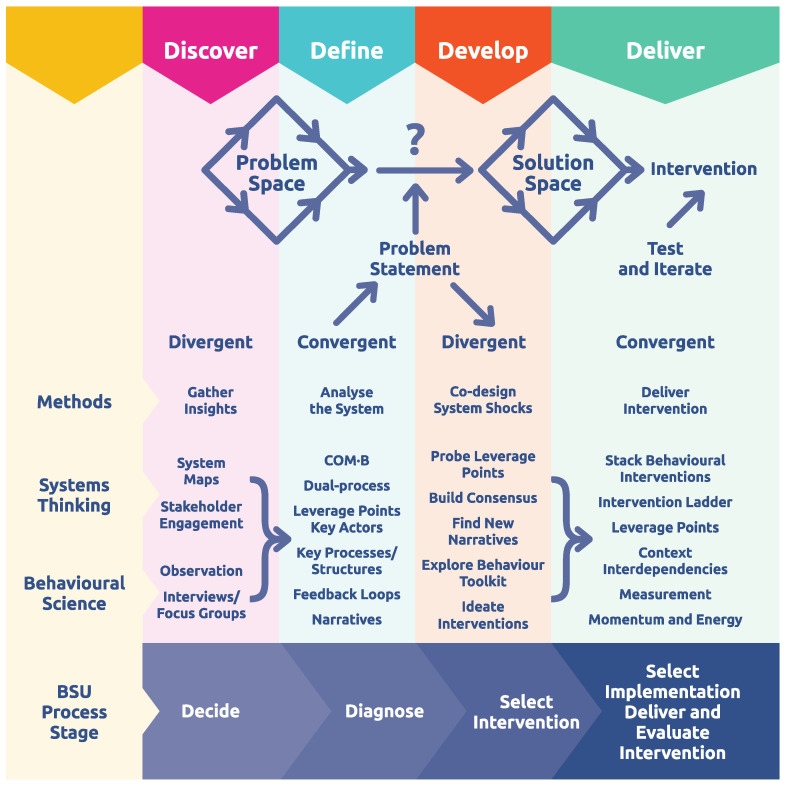
Integrating behavioural science and systems thinking through a design thinking structure.

**Figure 6 behavsci-15-00403-f006:**
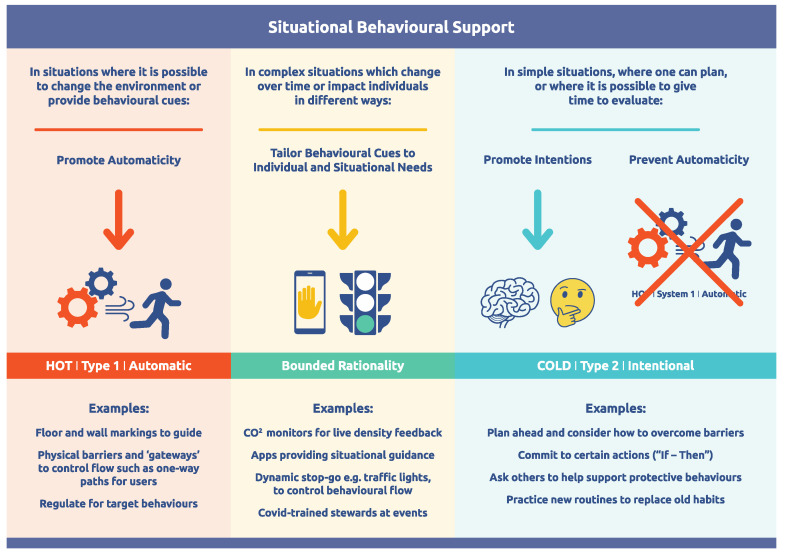
Adapted from [27] ([27]) *Behavioral Sciences*.

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
