# Peer review of "Integrating Systems Thinking and Behavioural Science"

_behavsci, 2025, doi:10.3390/bs15040403_

Round 1
Reviewer 1 Report
Comments and Suggestions for Authors
This manuscript was well refined, well written and composed. The authors have defined key terms and areas of research, demonstrated links between complex areas and applications and a thorough understanding of how the many components come together, interrelate, strengthen or take from each. I had no concerns or suggestions for the authors and would encourage publication.
Author Response
Reviewer 1 comments:
"This manuscript was well refined, well written and composed. The authors have defined key terms and areas of research, demonstrated links between complex areas and applications and a thorough understanding of how the many components come together, interrelate, strengthen or take from each. I had no concerns or suggestions for the authors and would encourage publication."
RESPONSE 1
We would like to thank the reviewer for their complimentary and encouraging comments. No additional actions have been taken.
Reviewer 2 Report
Comments and Suggestions for Authors
This is a thoughtful and interesting discussion. The thesis is well-defended and there is no shortage of evidence to support it. I have no criticisms of the conclusion. My only comments are simply responding to my own interest in the lack of practical application in the real world of these fairly well established ideas. In fact, I would say that the lack of discussion of cognitive biases in policy and general education about them is conspicuous in education, policy, and practice. I would love to hear the author's thoughts about this absence. In the West, during covid, we witnessed every possible error in approaches to communication, the exacerbation not mitigation of cognitive bias, and now in North America, a political attack on principled scientific thinking and policy across many domains of health, environment, and society. While system thinking and behavioural science can explain why specific actions reveal bias, there is no well established mechanism to stigmatize such error. I wonder (of the conclusion of this paper) if the mechanisms of social control and propaganda are capable of endorsing strategies to approach even wicked problems, if the result is a spill over behaviour of social agents into domains that require these same biases as mechanisms of control through manipulation or other means described Manufacturing Consent, for instance. Individuals tend to embrace analagous thinking/reasoning. If I'm shown the biases that affect my behaviour about alcohol consumption for instance, what is to prevent me from applying those new filters to my governments list of political allies or to my governments general behaviour. I reflect on how unwelcome David Nutt's hierarchy of substance harm was received by political officials as it demonstrated to anyone the inherent bias and irrational nature of policy towards stigmatized and non-stigmatized substances. Perhaps enabling a general public with the capacity to competently analysis the nature of any given risk is as dangerous to the structures of power as given the pubic weapons? In short, this paper reminded me that academics have understood the core findings of these recommendations for decades. The flaws in reasoning that this proposal might mitigate are responsible for millions of deaths per year. Why isn't this scientific work part of the core of Western education? Thanks for this excellent work.
Author Response
Reviewer 2 comments:
"This is a thoughtful and interesting discussion. The thesis is well-defended and there is no shortage of evidence to support it. I have no criticisms of the conclusion. My only comments are simply responding to my own interest in the lack of practical application in the real world of these fairly well established ideas. In fact, I would say that the lack of discussion of cognitive biases in policy and general education about them is conspicuous in education, policy, and practice. I would love to hear the author's thoughts about this absence. In the West, during covid, we witnessed every possible error in approaches to communication, the exacerbation not mitigation of cognitive bias, and now in North America, a political attack on principled scientific thinking and policy across many domains of health, environment, and society. While system thinking and behavioural science can explain why specific actions reveal bias, there is no well established mechanism to stigmatize such error. I wonder (of the conclusion of this paper) if the mechanisms of social control and propaganda are capable of endorsing strategies to approach even wicked problems, if the result is a spill over behaviour of social agents into domains that require these same biases as mechanisms of control through manipulation or other means described Manufacturing Consent, for instance. Individuals tend to embrace analagous thinking/reasoning. If I'm shown the biases that affect my behaviour about alcohol consumption for instance, what is to prevent me from applying those new filters to my governments list of political allies or to my governments general behaviour. I reflect on how unwelcome David Nutt's hierarchy of substance harm was received by political officials as it demonstrated to anyone the inherent bias and irrational nature of policy towards stigmatized and non-stigmatized substances. Perhaps enabling a general public with the capacity to competently analysis the nature of any given risk is as dangerous to the structures of power as given the pubic weapons? In short, this paper reminded me that academics have understood the core findings of these recommendations for decades. The flaws in reasoning that this proposal might mitigate are responsible for millions of deaths per year. Why isn't this scientific work part of the core of Western education? Thanks for this excellent work. "
RESPONSE 2
We wish to thank the reviewer for their supportive comments and their view of the significance and potential impact of this work.
In relation to our conclusion, the reviewer notes their own interest in the “spill over” of biases into political thinking. e.g. “... The lack of practical application in the real world of these fairly well established ideas.” And “... the lack of discussion of cognitive biases in policy and general education about them is conspicuous...” And finally “The flaws in reasoning that this proposal might mitigate are responsible for millions of deaths per year. Why isn't this scientific work part of the core of Western education? Thanks for this excellent work.”
We are encouraged that the reviewer highlights the significant practical and policy implications of this review. We share the reviewers views about the importance of building capability in the general public in understanding biases in thinking and behaviour, and the far-reaching impact that would have. As such, we have added content to the conclusion (section 10) to draw attention to this issue (lines 900-909). In adding this content, we also noticed one or two sentences which needed correcting and so have made minor non-substantive corrections to the conclusion section (lines 847, 856, and 874).
Reviewer 3 Report
Comments and Suggestions for Authors
The article presents a narrative review of the literature with the aim of presenting the integration of systems thinking with behavioral science to provide a novel framework for addressing complex societal issues that affect well-being. The authors present a strong, well-written article that effectively introduces and integrates concepts into a compelling narrative. I have just a few suggestions to provide more clarity.
Abstract:
- Clearly and concisely written; no suggestions for revision.
Section 1:
- Nicely summarizes the article; no suggestions for revision.
Section 2:
- While I thought this was clearly written, I felt that it is a little light on references, particularly in the second paragraph (Page 2, lines 78-90). There are several references at the end of that paragraph, but those focus only on dual-process theories.
Section 3:
- Beginning in this section, the concepts become complicated. This may be the concept that readers are least familiar with and most likely to get lost in the details. The authors state “one example that is depicted in Figure 1 and explored further below . . .” However, I was unable to connect the text following the figure to what was depicted in the figure, which was somewhat confusing. It wasn’t until Section 4 that I began to make some connections between the figure and the text. Standing alone, the Figure is not very informative. Perhaps moving it down to the section where boundaries, perspective, and interconnections are discussed may be helpful to guide the reader through these sections.
- In Box 1, the figure could be cleaned up a bit (remove the left border).
Section 4:
- See my note above about more explicitly tying Figure 1 to the text in this section.
- Figure 2a is so small that it is hard to even make out what it is supposed to represent. The zoomed-in section in Figure 2b is also difficult to read, even when I enlarge it to 200%. Consider making this figure larger and removing Figure 2a if the content isn’t necessary to understand the argument.
Section 5:
- No suggestions for revision
Section 6:
- Consider enlarging the font of the text within Figure 3, particularly the text in the boxes.
Section 7:
- No suggestions for revision
Section 8:
- No suggestions for revision
Section 9:
- Figure 5 does a nice job of pulling everything together and is the highlight of the paper, in my opinion. Could it be made slightly larger to make the font easier to read? Same for Figure 6.
Section 10:
Effectively summarizes and ties concepts together; no suggestions for revision.
References:
- There are many works cited that are more than 10 years old. However, these seem to be seminal articles that introduce theoretical frameworks and thus may be appropriate for the purpose. Consider sections where updated references may be relevant.
Author Response
Reviewer 3 comments:
"The article presents a narrative review of the literature with the aim of presenting the integration of systems thinking with behavioral science to provide a novel framework for addressing complex societal issues that affect well-being. The authors present a strong, well-written article that effectively introduces and integrates concepts into a compelling narrative. I have just a few suggestions to provide more clarity.
Abstract:
- Clearly and concisely written; no suggestions for revision.
Section 1:
- Nicely summarizes the article; no suggestions for revision.
Section 2:
- While I thought this was clearly written, I felt that it is a little light on references, particularly in the second paragraph (Page 2, lines 78-90). There are several references at the end of that paragraph, but those focus only on dual-process theories.
Section 3:
- Beginning in this section, the concepts become complicated. This may be the concept that readers are least familiar with and most likely to get lost in the details. The authors state “one example that is depicted in Figure 1 and explored further below . . .” However, I was unable to connect the text following the figure to what was depicted in the figure, which was somewhat confusing. It wasn’t until Section 4 that I began to make some connections between the figure and the text. Standing alone, the Figure is not very informative. Perhaps moving it down to the section where boundaries, perspective, and interconnections are discussed may be helpful to guide the reader through these sections.
- In Box 1, the figure could be cleaned up a bit (remove the left border).
Section 4:
- See my note above about more explicitly tying Figure 1 to the text in this section.
- Figure 2a is so small that it is hard to even make out what it is supposed to represent. The zoomed-in section in Figure 2b is also difficult to read, even when I enlarge it to 200%. Consider making this figure larger and removing Figure 2a if the content isn’t necessary to understand the argument.
Section 5:
- No suggestions for revision
Section 6:
- Consider enlarging the font of the text within Figure 3, particularly the text in the boxes.
Section 7:
- No suggestions for revision
Section 8:
- No suggestions for revision
Section 9:
- Figure 5 does a nice job of pulling everything together and is the highlight of the paper, in my opinion. Could it be made slightly larger to make the font easier to read? Same for Figure 6.
Section 10:
Effectively summarizes and ties concepts together; no suggestions for revision.
References:
•There are many works cited that are more than 10 years old. However, these seem to be seminal articles that introduce theoretical frameworks and thus may be appropriate for the purpose. Consider sections where updated references may be relevant."
RESPONSE 3
Firstly, we wish to thank the reviewer for the constructive suggestions, which we address below, and their acknowledge of the value of the work “The authors present a strong, well-written article that effectively introduces and integrates concepts into a compelling narrative.”
Further, the reviewer suggests some changes to various sections. Here we outline how we have resolved these recommendations:
Section 2: “While I thought this was clearly written, I felt that it is a little light on references, particularly in the second paragraph (Page 2, lines 78-90).”
We have added four references in this section of text to provide the additional references requested(lines 83, 87-88 and 94)
Section 3: “Standing alone, the Figure is not very informative. Perhaps moving it down to the section where boundaries, perspective, and interconnections are discussed may be helpful to guide the reader through these sections.”
We have moved Figure 1 as suggested and tied it into the text in section 4 (lines 298-303)
“In Box 1, the figure could be cleaned up a bit (remove the left border).”
We have tidied the figure. In fact, when I went to the Wiki page, there is a new version of the figure, updated by the original authors. So we have included that image as the replacement figure and referenced appropriately (lines 286-287).
Section 4: “See my note above about more explicitly tying Figure 1 to the text in this section.”
We have done this (line 305).
“Figure 2a is so small that it is hard to even make out what it is supposed to represent. The zoomed-in section in Figure 2b is also difficult to read, even when I enlarge it to 200%. Consider making this figure larger and removing Figure 2a if the content isn’t necessary to understand the argument.”
We have done this (lines 325-358)
Section 6: “Consider enlarging the font of the text within Figure 3, particularly the text in the boxes.”
We have enlarged the entire figure so that the font size in the figure is similar to the text font size of the manuscript (603-606).
Section 9: “Figure 5 does a nice job of pulling everything together and is the highlight of the paper, in my opinion. Could it be made slightly larger to make the font easier to read? Same for Figure 6.”
Likewise, we have enlarged the entire figure so that the font size in the figure is similar to the text font size of the manuscript (lines 747-749)
References: “There are many works cited that are more than 10 years old. However, these seem to be seminal articles that introduce theoretical frameworks and thus may be appropriate for the purpose. Consider sections where updated references may be relevant.”
As the reviewer notes, the older references are ‘seminal’ to the history of behavioral science and systems thinking. We have considered the references throughout and feel that the ones we have cited are relevant and appropriate.